# ^18^F-FDG and ^11^C-Methionine PET/CT in Newly Diagnosed Multiple Myeloma Patients: Comparison of Volume-Based PET Biomarkers

**DOI:** 10.3390/cancers12041042

**Published:** 2020-04-23

**Authors:** Maria I Morales-Lozano, Oliver Viering, Samuel Samnick, Paula Rodriguez-Otero, Andreas K Buck, Maria Marcos-Jubilar, Leo Rasche, Elena Prieto, K Martin Kortüm, Jesus San-Miguel, Maria J. Garcia-Velloso, Constantin Lapa

**Affiliations:** 1Department of Nuclear Medicine, University Clinic of Navarra, Center of Applied Medical Research (CIMA), Navarra Institute for Health Research (IDISNA), 31008 Pamplona, Spain; mmlozano@unav.es (M.I.M.-L.); eprietoaz@unav.es (E.P.); 2Department of Nuclear Medicine, University Hospital Würzburg, 97080 Würzburg, Germany; oli4ing@googlemail.com (O.V.); samnick_S@ukw.de (S.S.); buck_a@ukw.de (A.K.B.); Constantin.Lapa@uk-augsburg.de (C.L.); 3Department of Hematology, University Clinic of Navarra, CIMA, CIBERONC, IDISNA, 31008 Pamplona, Spain; paurodriguez@unav.es (P.R.-O.); mmarcos.3@unav.es (M.M.-J.); sanmiguel@unav.es (J.S.-M.); 4Department of Internal Medicine II, University Hospital Würzburg, 97080 Würzburg, Germany; rasche_l@ukw.de (L.R.); Kortuem_M@ukw.de (K.M.K.); 5Nuclear Medicine, Medical Faculty, University of Augsburg, Stenglinstrasse 2, 86156 Augsburg, Germany

**Keywords:** multiple myeloma, methionine, total lesion glycolysis (TLG), metabolic tumor volume (MTV), total lesion methionine uptake (TLMU)

## Abstract

^11^C-methionine (^11^C-MET) is a new positron emission tomography (PET) tracer for the assessment of disease activity in multiple myeloma (MM) patients, with preliminary data suggesting higher sensitivity and specificity than ^18^F-fluorodeoxyglucose (^18^F-FDG). However, the value of tumor burden biomarkers has yet to be investigated. Our goals were to corroborate the superiority of ^11^C-MET for MM staging and to compare its suitability for the assessment of metabolic tumor burden biomarkers in comparison to ^18^F-FDG. Twenty-two patients with newly diagnosed, treatment-naïve symptomatic MM who had undergone ^11^C-MET and ^18^F-FDG PET/CT were evaluated. Standardized uptake values (SUV) were determined and compared with total metabolic tumor volume (TMTV) for both tracers: total lesion glycolysis (TLG) and total lesion ^11^C-MET uptake (TLMU). PET-derived values were compared to Revised International Staging System (R-ISS), cytogenetic, and serologic MM markers such as M component, beta 2 microglobulin (B2M), serum free light chains (FLC), albumin, and lactate dehydrogenase (LDH). In 11 patients (50%), ^11^C-MET detected more focal lesions (FL) than FDG (*p* < 0.01). SUVmax, SUVmean, SUVpeak, TMTV, and TLMU were also significantly higher in ^11^C-MET than in ^18^F-FDG (*p* < 0.05, respectively). ^11^C-MET PET biomarkers had a better correlation with tumor burden (bone marrow plasma cell infiltration, M component; *p* < 0.05 versus *p* = n.s. respectively). This pilot study suggests that ^11^C-MET PET/CT is a more sensitive marker for the assessment of myeloma tumor burden than ^18^F-FDG. Its implications for prognosis evaluation need further investigation.

## 1. Introduction

Multiple myeloma (MM) is a malignant disorder of clonal plasma cells (PCs) that represents approximately 10% of hematological malignancies [1]. Positron emission tomography (PET) in combination with computed tomography (CT) and ^18^F-fluorodeoxyglucose (^18^F-FDG) is a well-established imaging technique in MM. It has proven its value in the assessment of tumor burden and disease activity in MM, with sensitivity and specificity ranging from 80% to 100% [2,3]. Additionally, it has demonstrated prognostic impact, and the presence of more than three focal lesions (FL), maximum standardized uptake values (SUVmax) higher than 4.2, or the presence of extramedullary disease are, indeed, predictors of poor prognosis [4,5]. On top of this, post-therapy imaging studies have demonstrated that PET negativity after induction treatment [6,7] or autologous/allogeneic stem cell transplantation (ASCT) [7,8,9,10] correlates with longer progression-free survival (PFS). Moreover, new parameters measuring tumor burden in ^18^F-FDG PET/CT such as metabolic tumor volume (MTV) and total lesion glycolysis (TLG) have also demonstrated an association with survival in MM patients [11,12]. Furthermore, patients with high TLG had poor outcome even in the context of molecularly defined low-risk disease, highlighting the additional value of novel markers in the assessment of MM.

Nevertheless, ^18^F-FDG PET/CT is associated with several shortcomings: (1) the sensitivity is poor in patients with diffuse bone marrow (BM) infiltration; (2) ^18^F-FDG avidity may be absent in patients with low hexokinase-2 expression [13] and specificity is not reliable in inflammatory or infectious lesions. For these reasons, alternative tracers such as ^11^C-methionine (^11^C-MET) [14], ^11^C-choline [15], or radiolabeled C-X-C chemokine receptor 4 ligands [16,17] have been investigated. Several studies have suggested a higher sensitivity of ^11^C-MET PET/CT over ^18^F-FDG PET/CT in the detection of both intra- and extramedullary disease [18,19]. In addition, a better correlation of ^11^C-MET uptake with the degree of BM infiltration by malignant PCs has been recently reported [20]. 

The aim of the study was to assess the diagnostic value of volume-based ^11^C-MET PET/CT biomarkers in newly diagnosed, treatment-naïve MM patients in comparison to ^18^F-FDG.

## 2. Results

### 2.1. Patient-Based Analysis

On a patient basis analysis, both techniques were positive in all patients. In ^11^C-MET PET/CT, diffuse BM infiltration, as a single abnormality, was detected in 6/22 (27.3%) patients, while FL only were present in 3/22 cases (13.6%), and a combined pattern (both focal and diffuse) was detected in 13/22 patients (59.1%). With ^18^F-FDG PET/CT, disease was categorized as only diffuse BM in 6/22 (27.3%), FL in 10/22 (45.5%), and a combined pattern in 6/22 (27.3%) patients. These results indicate that ^11^C-MET PET/CT was able to detect combined infiltration in more patients than ^18^F-FDG PET/CT (13 versus 6 cases, respectively) (Figure 1), while ^18^F-FDG detected FL in more patients than ^11^C-MET (*p* = 0.003). No extramedullary disease was identified. An incidental finding consisting of a focal uptake on the right lobe of the prostate was only detected by ^18^F-FDG PET/CT and turned out to be an adenocarcinoma (Figure 1).

### 2.2. Lesion-Based Analysis

On a lesion basis analysis, ^18^F-FDG and ^11^C-MET were concordant, depicting > 3 FL in seven patients, < 3 FL in five patients, and diffuse infiltration in five patients (kappa agreement index = 0.66) (Table 1). However, more FL were detected by ^11^C-MET in 11 patients (50%, *p* < 0.01), 3 patients with FL, and 8 patients with combined infiltration (Figure 2). By contrast, ^18^F-FDG detected more FL in only one patient.

### 2.3. PET-Derived Biomarkers

We analyzed SUVmax, SUVmean, and SUVpeak for both techniques. Median, inter-quartile range (IQR), and SUVmax values were significantly higher for ^11^C-MET (16.40 (6–195.6)) than for ^18^F-FDG PET/CT (8.76 (3.45–62.23)) (*p* = 0.008). The same difference was observed for median SUVmean (^11^C-MET: 4.59 (2.79–8.35) versus ^18^F-FDG: 3.55 (1.82–7.74), *p* = 0.022 and SUVpeak (^11^C-MET: 10.72 (4.64–126.50) versus ^18^F-FDG: 6.56 (2.82–39.85), *p* = 0.005. 

Regarding volume-based biomarkers, median total metabolic tumor volume (TMTV) for ^11^C-MET PET/CT (443.4 cm^3^ (145.2–1102.6)) was significantly higher than median TMTV for ^18^F-FDG (134.8 cm^3^ (5.6–524.9); *p* < 0.001) with a median difference of 141.2% (21.6–18,369). The same finding was observed for total lesion ^11^C-MET uptake (TLMU) (^11^C-MET, median: 2021.4 g (761.6–6061.4) versus TLG ^18^F-FDG, median: 598.4 g, (10.7–2086.4), *p* < 0.001) with a median difference of 216.7% (40.3–19,212.2).

Global comparisons of TMTV and TLG/TLMU between ^11^C-MET and ^18^F-FDG PET/CT, as well as differences for each tracer depending on uptake patterns are shown in Figure 3. Differences in TMTV and TLG/TLMU between the two tracers on a patient’s based analysis are shown in Figure 4.

### 2.4. Correlation of PET Biomarkers with Tumor-Burden derived Parameters

We have analyzed the correlations between four different PET parameters (presence of FL, SUVmax, SUVmean, and volumetric parameters: TMTV and TLG/TLMU) and different clinical variables that reflect myeloma burden (M-component, percentage of plasma cells (PC), beta 2 microglobulin (B2M), free light chains (FLC) levels, creatinine, albumin, and Revised International Staging System (R-ISS).

No correlation was found between the number of FL detected by ^18^F-FDG or ^11^C-MET PET/CT and the M component (*p* = 0.71 for ^18^F-FDG PET and *p* = 0.81 for ^11^C-MET PET), B2M (*p* = 0.38 for ^18^F-FDG PET and *p* = 0.17 for ^11^C-MET PET), FLC levels (*p* = 0.24 for ^18^F-FDG PET and *p* = 0.82 for ^11^C-MET PET) nor R-ISS (*p* = 0.35 for ^18^F-FDG PET and *p* = 0.76 for ^11^C-MET PET).

Upon analyzing semi-quantitative parameters (SUVmax and SUVmean) obtained by ^18^F-FDG PET/CT, a moderate correlation was found with both SUVmax and SUVmean and high β2M levels (>3.5 mg/L) (SUVmax: *rs* = –0.458, *p* = 0.03; SUVmean: *rs* = –0.417; *p* = 0.05). However, no significant correlation was detected between SUVmean and SUVmax in ^18^F-FDG and albumin < 3.5 g/dL (*p* = 0.12 for SUVmax and *p* = 0.19 for SUVmean) or creatinine > 2 mg/dL (*p* = 0.66 for SUVmax, *p* = 0.44 for SUVmean).

In ^11^C-MET PET/CT, no correlation was identified between SUVmax and SUVmean and albumin < 3.5 g/dL (*p* = 0.38 for SUVmax and *p* = 0.43 for SUVmean), creatinine > 2 mg/dL (*p* = 0.74 for SUVmax, *p* = 0.44 for SUVmean), or β2M > 3.5 mg/L (*p* = 0.29 for SUVmax and *p* = 0.64 for SUVmean). In addition, no differences were present between SUVmax values and cytogenetic risk in ^18^F-FDG PET/CT nor ^11^C-MET PET/CT (*p* = 0.32 for ^18^F-FDG and *p* = 0.66 for ^11^C-MET).

Finally, we explored correlations between volumetric parameters (TMTV and TLG/TLMU values) obtained with both tracers and clinical variables. A positive correlation was found between TMTV and B2M levels both in ^18^F-FDG PET/CT (*r* = 0.434, *p* = 0.044) and in ^11^C-MET PET/CT (*r* = 0.569, *p* = 0.006), in which the latter had a higher correlation. Moreover, in ^11^C-MET PET/CT, significant correlations between TMTV and the M-component (*r* = 0.781, *p* = 0.003) or BM infiltration (0.571, *p* = 0.007) were recorded. With respect to TLG, no correlations were detected in ^18^F-FDG PET/CT, whereas moderate to good positive correlations were demonstrated in ^11^C-MET PET/CT for TLMU and B2M levels (*r* = 0.428, *p* = 0.047), TLMU, M component (*r* = 0.616, *p* = 0.033), and TLMU and BM infiltration (*r* = 0.450, *p* = 0.041). No other correlations were present for the rest of the serum parameters explored.

## 3. Discussion

The use of PET-CT has rapidly expanded in MM both for the evaluation of disease extension at presentation (particularly for detection of extramedullary disease) as well as for treatment response assessment [21]. Although ^18^F-FDG is the gold standard radiotracer, recent data suggest that ^11^C-MET may be a potentially superior radiotracer for MM imaging due to higher sensitivity than ^18^F-FDG [22].

The current study conducted in patients with newly diagnosed, treatment-naïve MM confirms the higher sensitivity of ^11^C-MET PET/CT, with about 50% of subjects demonstrating more lesions as compared to ^18^F-FDG PET/CT, thus being in line with previous preliminary observations [18,19,20,22]. Furthermore, we have investigated for the first time ^11^C-MET PET volume-based biomarkers (TMTV, TMLU) demonstrating a positive correlation with MM serum markers of tumor burden such as M component, bone marrow (BM) infiltration, or B2M. In contrast, ^18^F-FDG derived markers only showed a modest correlation with ß2M, while no association was found with other indicators of disease including albumin, FLC, M component, or the percentage of BM infiltration by malignant plasma cells (assessed by random iliac crest BM biopsy). Thus, ^11^C-MET might be considered a highly sensitive radiotracer to be used as a non-invasive surrogate for the investigation of MM disease activity, including the detection of minimal residual disease.

On the other hand, ^18^F-FDG PET/CT remains the standard technique for the nuclear medicine-based evaluation of myeloma, given its availability and the vast body of experience in various clinical settings. Beyond mere diagnostic accuracy, ^18^F-FDG has proven its prognostic value in various studies and has even outperformed magnetic resonance imaging in terms of therapy monitoring and response assessment [7]. In addition, recent studies with ^18^F-FDG PET/CT have investigated the prognostic value of the new metabolic biomarkers (TMTV and TLG) and have suggested significant survival implications at baseline and a more precise quantitation of the glycolytic phenotype of active disease [11,12]. The aim of the current study was just to compare the performance of C-MET versus F-FDG in newly diagnosed MM patients, and it was out of the scope to explore the prognostic value of C-MET due to the small sample size, retrospective nature, short follow up, and treatment heterogeneity. However, as mentioned above, it is important to underscore the correlation between volume-based biomarkers and MM tumor burden markers (both median TMTV and TLMU), which were clearly higher in ^11^C-MET PET/CT as compared to ^18^F-FDG PET/CT. Whether this would translate in better survival prognostication and individually tailored treatment decisions remains to be investigated. It is necessary to take into account that the short half-life of C-11 makes necessary an on-site cyclotron, which could be a major limitation for the tracer’s widespread use.

In addition, there is still a need for the standardization of segmentation methods when calculating TMTV. In the present study, it was decided to segment individually by an absolute or a relative threshold instead of employing a global threshold of the CT image, as described by Takahashi et al. [23] or a fixed threshold of 40% of SUVmax as described by Fonti et al. [24]. In our experience, the use of a fixed cut-off value does not necessarily work for all patients, especially when focal disease is present or double tracer studies are carried out (Appendix A). Our proposal is to use a relative SUVmax threshold (i.e., SUV > 41% of SUVmax) whenever diffuse or combined uptake is present, while in the presence of FL only, results are not unequivocal and further studies are needed to achieve a general optimal single threshold. 

Despite all limitations mentioned above, this two-center study is the first that has investigated ^11^C-MET PET volume-based biomarkers (TMTV, TLMU) and demonstrated a correlation between these new ^11^C-MET-PET biomarkers and other MM prognostic factors [14].

## 4. Materials and Methods

### 4.1. Study Description and Patient Population

The study has been approved by the University Hospital of Würzburg (212/13) and by the University Clinic of Navarra (161/2015) ethics committees. All patients signed an informed consent form according to the Declaration of Helsinki. ^11^C-MET was administered under the conditions of the German and Spanish pharmaceutical law (German Medicinal Products Act, AMG §13 2b; RD 1015/2009) and in accordance with the responsible regulatory bodies (Regierung von Oberfranken, Germany; AEMPS, Spain).

Twenty-two consecutive patients with newly diagnosed, treatment-naïve MM referred for dual tracer (^18^F-FDG and ^11^C-MET) staging PET/CT were retrospectively reviewed by PET experts from the University Hospital of Würzburg (C.L., Würzburg, Germany) and University Clinic of Navarra (M.J.G.-V., Pamplona, Spain). None of the patients have been previously reported in previous publications. Patients with plasma cell malignancies other than MM (e.g., smoldering myeloma) were excluded from the analysis. The following characteristics were documented and subsequently analyzed: age, gender, hemoglobin, calcium, serum creatinine, C-reactive protein, B2M, albumin, M component, percentage of malignant plasma cell infiltration as assessed by random bone marrow biopsy of the iliac crest, lactate dehydrogenase (LDH), platelet count, type of monoclonal component, presence and level of serum light chains, clinical staging according to the R-ISS, and chromosomal abnormalities defining high-risk patients: t(4;14) and /or t(14;16) and/or del(17p). Patients´ characteristics are summarized in Table 2 and Appendix A.

### 4.2. PET/CT Acquisition

^18^F-FDG and ^11^C-MET were synthesized in-house with a 16 MeV Cyclotron (Würzburg; GE PET trace 6; GE Healthcare, Milwaukee, USA) or an 18 MeV Cyclotron (Navarra; Cyclone 18/9, IBA Radiopharma Solutions, Belgium). PET/CT was performed in both institutions on a PET/CT scanner (Siemens Biograph mCT 64, Siemens, Knoxville, USA) within a median interval of 1 day between ^18^F-FDG and ^11^C-MET scans (range, 0–11).

Patients fasted at least 4 hours before ^18^F-FDG (3 to 5 MBq/kg) and ^11^C-MET injection (6–10 MBq/kg). No adverse effects associated to radiotracer injection were observed. PET/CT scans were acquired after 60 min (8F-FDG) or 20 min (^11^C-MET), using contrast-enhanced CT with dose modulation and a quality reference of 210 mAs (Würzburg)) or non-contrast-enhanced CT with Care Dose 4D and a quality reference of 80–120 mAs (Würzburg, Navarra), including the skull to the proximal thighs and lower limbs. Consecutively, PET emission data were acquired in 3D-mode with 2 min (Würzburg) or 2–3 min (Navarra) emission time per bed position in the skull to mid-thighs and 1 min/bed in the lower limbs. After decay and scatter correction, PET data were reconstructed according to standard protocols consisting of 3D ordinary Poisson ordered-subset expectation maximization (OSEM) iterative reconstruction with time-of-flight and point spread function modeling, 3 iterations, and 21 subsets, a 2 mm full-width at half-maximum Gaussian post-filter and a 200 × 200 image matrix. 

### 4.3. PET/CT Assessment

Two experienced nuclear medicine physicians (M.J.G.-V. and C.L.) blinded to the results of clinical and biologic data visually assessed PET/CT anonymized images. ^18^F-FDG PET maximum intensity projection (MIP) and axial/sagittal/coronal images were reviewed and focal lesions as well as bone infiltration were characterized following the patterns defined by Moreau et al [7].

For ^11^C-MET, every focal uptake with higher activity than the surrounding normal tissue or contralateral structure was considered positive. Criteria for the diagnosis of involvement of BM was focally increased ^11^C-methionine uptake in the BM or diffusely increased ^11^C-methionine in the whole hematopoietic BM with or without the expansion of BM into distal parts of long bones [14]. BM biopsy, performed without clinical information nor PET/CT results, served as the standard of reference in all cases.

Thereafter, ^18^F-FDG and ^11^C-MET PET/CT images underwent a three-dimensional volume of interest analysis of the axial and appendicular skeleton with “PET/CT Viewer Beth Israel for FIJI” [25,26,27]. This software allows calculating SUVmax, SUVmean and SUVpeak values as well as the new biomarkers MTV, which were defined for each lesion as the sum of voxels exceeding an absolute SUV threshold or relative threshold, and the TLG, which was calculated with the following formula: TLG = ∑ (SUVmean × MTV). For ^11^C-MET PET, the equivalent term for TLG was the total lesion methionine uptake (TLMU), which was defined as MTV times the mean standardized uptake values (SUVmean) within the boundary [28]. The total MTV (TMTV) in each patient was defined as the sum of MTV of all the individual lesions obtained.

For the calculation of tumor volume biomarkers, the software automatically delineated the tumor volume, requiring operator supervision to discard those physiological uptakes (bladder, brain, liver, etc.). Later, a threshold was set so that the software detected all voxels included in that cut-off point. In addition to the 41% of SUVmax threshold (calculated at the local maximum point) recommended by the European Association of Nuclear Medicine [29], other absolute (SUV > 1, 3, and 4) and relative thresholds (SUV > 30% and 50% of SUVmax) were used to segment the images (Appendix A). Finally, the automatically generated thresholds were evaluated, and the threshold that best fit with the visually identified active lesions was chosen. In ^18^F-FDG PET/CT, the best threshold for segmentation was SUV > 41% of SUVmax in 15/22 patients (68.2%), while in other patients, different thresholds were eligible because of under- or overestimation. The second most selected threshold was SUV > 4 in 4/22 patients (18.2%), and the remaining cases corresponded to a threshold of SUV > 50% SUVmax in 2/22 patients (9.1%) and SUV >1 in 1 patient (4.5%). An example of these different criteria is demonstrated in Appendix A. In ^11^C-MET PET/CT, similar results were obtained, with a relative threshold SUV > 41% SUVmax as the most frequently selected (16/22, 72.7%), and some cases with different thresholds because of under- or overestimation, SUV > 30% SUVmax (1/22, 4.5%), SUV > 50% SUVmax (1/22, 4.5%), and SUV > 4 (4/22, 18.2%). Overall, in 10 patients with focal disease in ^18^F-FDG, different segmentation methods on FIJI software were employed and are shown in Appendix A. Appendix A shows a representative image of the impaired results of selecting a fixed threshold of SUV > 41% SUVmax in the case of focal disease. Interestingly, when a diffuse or combined focal/diffuse pattern was present, only relative thresholds fit better with the disease extension in all cases. This pattern was also present for ^11^C-MET PET/CT with the exception of two cases with combined infiltration in which SUV > 4 was preferred.

### 4.4. Statistical Analysis

Quantitative data are presented as the median, inter-quartile range (IQR), and mean ± SD, as appropriate. Spearman correlation was used to estimate linear relationships. A Chi square or Fisher exact test was conducted for comparison of frequency data between independent subgroups. The Wilcoxon test was the non-parametric statistical test used to compare two related samples, matched samples, or repeated measurements. A comparison of quantitative values from three independent groups was performed using the Kruskal–Wallis test. Concordance between ^11^C-MET and ^18^F-FDG PET imaging was presented in a two-way table. The level of agreement between the two evaluations was expressed by kappa statistics. Correlations between categorical and quantitative variables were performed using Spearman’s Rho.

Statistical analyses were performed in R (version 3.4.4, R Core Team, Vienna, Austria, 2018) and SPSS (version 22.0; SPSS, Inc. Chicago, IL, USA). All statistical tests were performed two-sided, and a *p*-value < 0.05 was considered to indicate statistical significance.

## 5. Conclusions

In summary, our preliminary results show that ^11^C-MET seems to be a more sensitive and accurate surrogate for total myeloma burden as compared to ^18^F-FDG. Our results might stimulate future research and be considered as groundwork for future prospective studies with larger sample sizes.

## Figures and Tables

**Figure 1 cancers-12-01042-f001:**
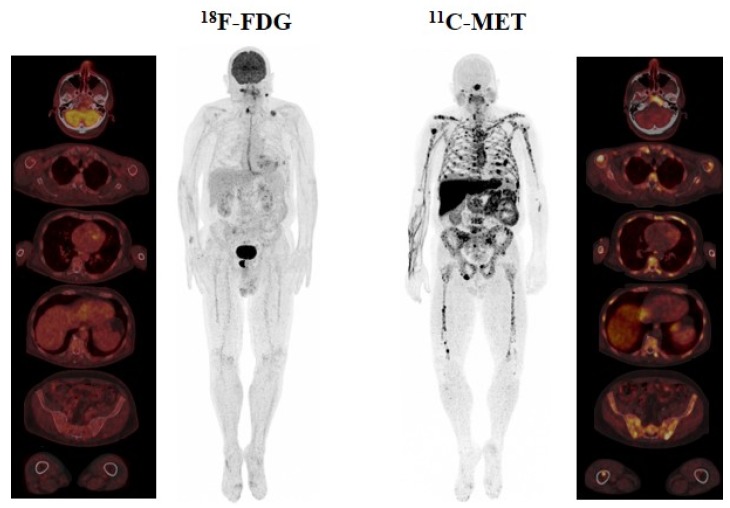
Display of a 68-year-old male with Bence–Jones kappa multiple myeloma Revised International Staging System (R-ISS) II. In ^18^F-fluorodeoxyglucose (^18^F-FDG) positron emission tomography (PET)/computed tomography (CT), focal lesions (FL) located on the right thyroid cartilage and sacral bone were identified. Nevertheless, the focal uptake on the lower cervical region and left clavicle corresponded to a lymph node and a fracture, respectively, and were thus considered unspecific findings. Please note that diffuse uptake in ^18^F-FDG PET/CT was homogeneous and below the liver cut-off. Conversely, a combined infiltration pattern with bone marrow (BM) infiltration and more than 10 FL was present in ^11^C-MET PET/CT. Importantly, they did not demonstrate a correlation in ^18^F-FDG PET/CT (fusion images).

**Figure 2 cancers-12-01042-f002:**
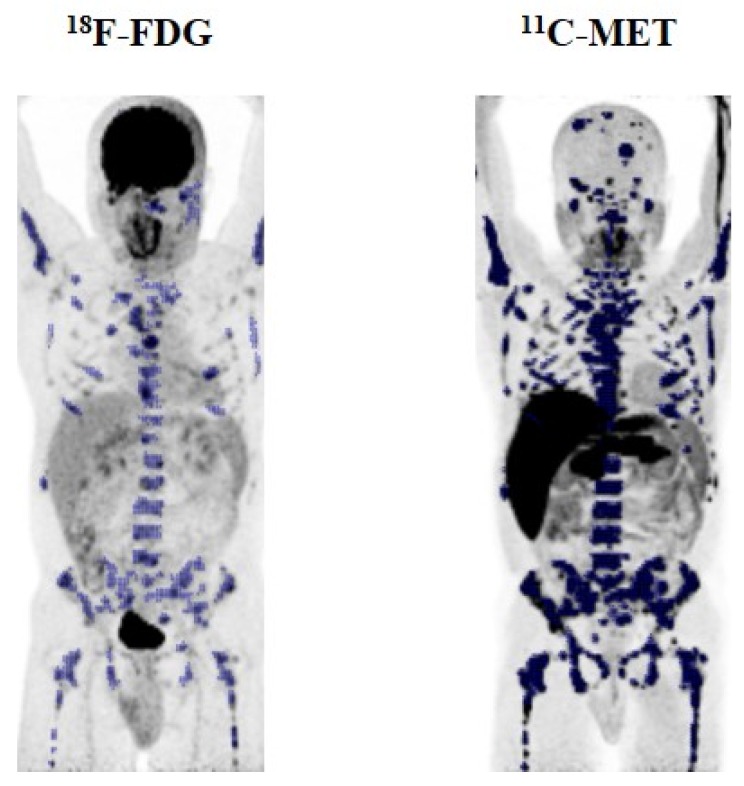
Display of a 47-year-old male (patient #10) with multiple myeloma R-ISS II. A combined pattern of diffuse and focal disease and more than three FL were detected by ^18^F-FDG PET/CT. The same lesions were detected by ^11^C-MET PET/CT and, noticeably more FL were detected, some of them located in the skull, leading to a remarkable difference of 103.3% in total metabolic tumor volume (TMTV) (TMTV MET: 726 cm^3^ vs. TMTV FDG: 357.1) and 366.1% in TLG/TLMU (MET TLMU: 6061.4 g vs. FDG TLG: 1300.4 g).

**Figure 3 cancers-12-01042-f003:**
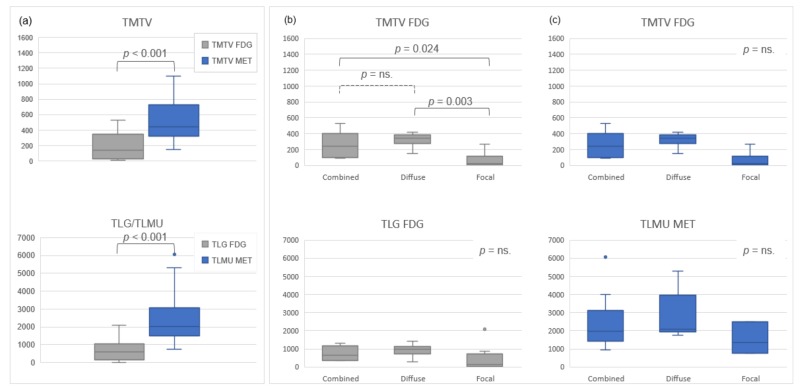
Global comparison of TMTV and total lesion glycolysis (TLG)/total lesion ^11^C-MET uptake (TLMU) between ^11^C-MET and ^18^F-FDG PET/CT scan (**a)**, and comparison of TMTV and TLG/TLMU, according to the observed pattern, for ^18^F-FDG (**b**) and ^11^C-MET (**c**).

**Figure 4 cancers-12-01042-f004:**
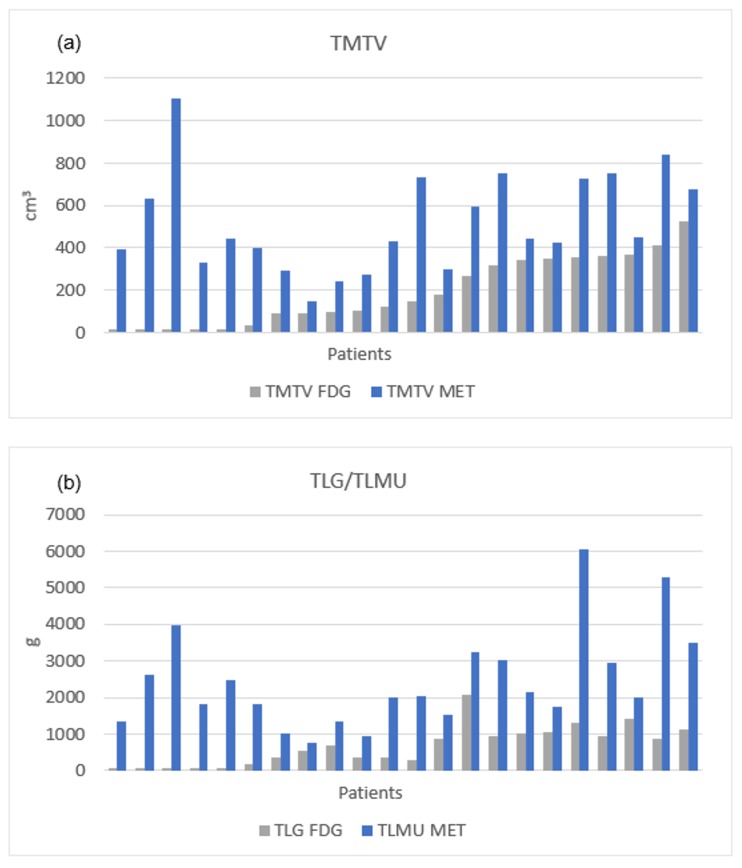
Differences in TMTV (**a**) and TLG/TLMU (**b**) between the two tracers on a patient-based analysis sorted by tumor burden in ^18^F-FDG PET/CT.

**Table 1 cancers-12-01042-t001:** Comparison of diagnostic performance for ^18^F-FDG and ^11^C-methionine (^11^C-MET) PET/CT.

PET/CT Results		MET-PET				Total
		Focal, >3 FL	Combined, <3 FL	Combined, >3 FL	Diffuse	
FDG-PET	Focal, <3 FL	0	1	3	0	4
	Focal, >3 FL	3	0	3	0	6
	Combined, <3 FL	0	4	0	1	5
	Combined, >3 FL	0	0	1	0	1
	Diffuse	0	1	0	5	6
Total		3	6	7	6	22

**Table 2 cancers-12-01042-t002:** Patients’ characteristics.

No.	Sex	Age	Myeloma Type	R-ISS	High-Risk Cytogenetics
1	male	75	IgG kappa	n/a	n/a
2	male	64	IgA kappa	Stage II	yes
3	male	54	IgA kappa	Stage II	no
4	female	56	IgA lambda	Stage II	n/a
5	male	59	kappa	Stage I	no
6	male	48	IgG kappa	Stage I	no
7	female	74	IgG lambda	Stage II	no
8	female	62	IgG kappa	Stage I	no
9	male	63	kappa	Stage III	yes
10	male	47	kappa	Stage II	no
11	male	59	IgG kappa	Stage I	no
12	male	72	IgG lambda	Stage II	no
13	male	61	kappa	Stage II	yes
14	male	68	IgG kappa	Stage II	yes
15	male	61	IgG kappa	Stage I	no
16	female	61	IgG kappa	Stage II	yes
17	male	37	kappa	Stage I	no
18	male	79	IgG kappa	n/a	no
19	male	68	kappa	Stage II	no
20	female	43	IgG lambda	Stage I	no
21	male	46	lambda	Stage II	yes
22	female	44	IgG kappa	Stage II	n/a

n/a = not available; high-risk cytogenetics is defined as t (4;14) and/or t (14;16) and/or del(17p); Ig: Immunoglobulin; R-ISS = Revised International Staging System.

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
