# Peer review of "18F-FDG and 11C-Methionine PET/CT in Newly Diagnosed Multiple Myeloma Patients: Comparison of Volume-Based PET Biomarkers"

_cancers, 2020, doi:10.3390/cancers12041042_

Round 1
Reviewer 1 Report
Lozano et al., in this study performed a direct comparison of the 11C-Met PET/CT scan to that of the extensively used 18G-FDg technique. The conclusions that the author drew from their studies is optimal based on their findings. The background information is quite extensive and to the point and therefore would grasps readers and medical practitioners attention. This paper marks as a starting point to study efficacy of the new technique and how it can be used extensively in the future.
Author Response
We deeply thank the reviewer for his/her thorough review and the interest in our study.

Reviewer 2 Report
In this manuscript Morales-Lozano et al. conducted a phase 2 trial in order to evaluate the superiority of 11C-MET for MM staging and to compare its suitability for assessment of metabolic tumor burden biomarkers in comparison to 18F-FDG in myeloma patients. The authors analyzed twenty-two patients with newly diagnosed, treatment-naïve 22 symptomatic MM who had undergone 11C-MET and 18F-FDG PET/CT were evaluated. Standardized 23 uptake values (SUV) were determined and compared with total metabolic tumor volume (TMTV) for both tracers, total lesion glycolysis (TLG) and total lesion 11C-MET uptake (TLMU). Confirming previous data they found that 50% of subjects demonstrating more lesions with 11C-MET PET/CT detects compared to 18F-FDG PET/CT. Furthermore, they are showing for the first time that 11C-MET PET volume-based biomarkers demonstrated a positive correlation with MM serum markers of tumor burden such as M component, bone marrow (BM) infiltration or B2M. In contrast, 18F-FDG derived markers that only showed a modest correlation with ß2M, while no association was found with other indicators of disease including percentage of BM infiltration by malignant.
In conclusion this is a very well written and scientifically sound manuscript. The reported data have been accurately analyzed and reported. I strongly recommend this manuscript for publication.
Author Response

(The authors gave the same response as above.)

Reviewer 3 Report
This is a retrospective comparison of the two tracers, a standard (FDG) and a new one (11C-MET). The study is small with only 22 patients but still the information provided is significant.
Some comments can be found below :
Please clarify that imaging with both tracers was performed before start of therapy (including dexamethasone) in all cases.
Please clarify the selection criteria: the authors state that 22 consecutive patients with newly diagnosed, treatment-naïve MM were referred for dual tracer staging PET/CT and then were retrospectively reviewed by PET experts. Is this a prospective or retrospective study?
The comparisons in a small number of patients (N=22) result in a highly significant p-value when groups have 6/22 vs 13/22 patients. Can you provide chi-square values?
11C-MET detects more lesions on a case by case comparison: are the lesions detected the same with both tracers in the same patient ? In figure 2 for example looks that the lesions depicted are the same
Why was the cutoff for comparison set to >3 lesions ?
In correlation analysis a comparison per ISS stage should also be included rather than with albumin and b2microglobulin alone
One of the sites of extramedullary disease is liver. Could 11C-MET identify any lesions in the liver? Were any patients with extramedullary disease included in the study ? Is 11C-MET sensitive for extramedullary lesion detection?
The median interval between 18F-FDG and 11C-MET scans was 1 day (range, 0-11); could this short distance between the two affect the result?
In the methods section , for 11C-MET evaluation is stated that “BM biopsy, performed without clinical information nor PET/CT results, served as standard of reference in all cases.”; Can you clarify how this works as a reference for the nuclear technique, given the patchy marrow infiltration ?
Did the patients have data on spine MRI? This technique is rather sensitive in detecting BM diffuse infiltration and could be used as a reference
Do the authors have any data on the 11C-MET uptake after therapy?
Author Response
This is a retrospective comparison of the two tracers, a standard (FDG) and a new one (11C-MET). The study is small with only 22 patients but still the information provided is significant.
Response: We deeply thank the reviewer for his/her thorough review and the interest in our study.
Some comments can be found below:
Please clarify that imaging with both tracers was performed before start of therapy (including dexamethasone) in all cases.
Response: We can clarify that imaging with both tracers was performed before start of any therapy. This piece of information can now be found in line 235 on page 8/16.
Please clarify the selection criteria: the authors state that 22 consecutive patients with newly diagnosed, treatment-naïve MM were referred for dual tracer staging PET/CT and then were retrospectively reviewed by PET experts. Is this a prospective or retrospective study?
Response: At our institutions, both FDG- and MET-PET/CT are routinely performed prior to treatment initiation. The analysis was performed retrospectively. This information is available in line 206 on page 7/16 and in line 235 on page 8/16.
The comparisons in a small number of patients (N=22) result in a highly significant p-value when groups have 6/22 vs 13/22 patients. Can you provide chi-square values?
Response: We thank the reviewer for raising this important point. We have reviewed the statistical analyses and the correct chi-square p-value is 0.003. This information has been corrected in line 74 on page 2/16.
11C-MET detects more lesions on a case by case comparison: are the lesions detected the same with both tracers in the same patient? In figure 2 for example looks that the lesions depicted are the same
Response: We thank the reviewer for raising this important point. Indeed, 11C-MET detected the same lesions than FDG in every patient but it also detected more lesions not identified with FDG. The legend in Figure 2 has been modified to emphasize that important point (line 94, page 4/16).
Why was the cutoff for comparison set to >3 lesions?
Response: The cut-off was set to three focal lesions as most prospective studies have established presence of three focal lesions to be an independent negative prognostic factor: Bartel TB et al 2009 (reference 4), Zamagni E et al 2011 (reference 5).
In correlation analysis a comparison per ISS stage should also be included rather than with albumin and b2microglobulin alone.
Response: This is a very good suggestion. We implemented respective R-ISS stages in the revised version of our manuscript. This piece of information can now be found in line 158, page 6/16.
One of the sites of extramedullary disease is liver. Could 11C-MET identify any lesions in the liver? Were any patients with extramedullary disease included in the study? Is 11C-MET sensitive for extramedullary lesion detection?
Response: We totally agree with the reviewer that detection of myeloma liver involvement by MET-PET/CT is hampered by high physiologic hepatic tracer uptake and might be missed by analysis. In the current patient cohort, we could not detect any extramedullary lesions in any patient (line 74, page 2/16). In general, sensitivity of MET is also high for EMD (references 19 and 20).
The median interval between 18F-FDG and 11C-MET scans was 1 day (range, 0-11); could this short distance between the two affect the result?
Response: We thank the reviewer for this interesting remark. The very short intervals between both scans were intended to provide the best posible comparability between both tracers. Given the rather short half-lives of 18F and 11C and the need for five half-lives to rule out significant effects of the former examination, one day or even 4 hours intervals (for the sequence MET à FDG) should not result in relevant confounding background noise.
In the methods section, for 11C-MET evaluation is stated that “BM biopsy, performed without clinical information nor PET/CT results, served as standard of reference in all cases.”; Can you clarify how this works as a reference for the nuclear technique, given the patchy marrow infiltration?
Response: We thank the reviewer for this interesting remark. We totally agree with the reviewer that myeloma BM infiltration is patchy and imaging can provide important information in this regard. However, BM biopsies are routinely performed by random aspiration of the posterior iliac crest and are thus the standard of reference in most centers of the world.
Did the patients have data on spine MRI? This technique is rather sensitive in detecting BM diffuse infiltration and could be used as a reference
Response: We totally agree with the reviewer that MRI is a very useful and sensitive tehcnique in the detection of myeloma, especially in the setting of diffuse BM involvement. Unfortunately, this technique was not available in all patients and was beyond the scope of this research project.
Do the authors have any data on the 11C-MET uptake after therapy?
Response: Unfortunately, we cannot provide this data.

Round 2
Reviewer 3 Report
I have no further comments